

# Impact of waves on phytoplankton activity on the Northwest European Shelf: insights from observations and km-scale coupled models

Dale Partridge[1,4,*], Ségolène Berthou[2], Rebecca Millington[1], James R. Clark[1], Lucy Bricheno[3], Juan Manuel Castillo[2], Julia Rulent[3], and Huw Lewis[2]

[1]Plymouth Marine Laboratory, Plymouth, PL1 2LP, UK
[2]Met Office, Exeter, EX1 3PB, UK
[3]National Oceanography Centre, Liverpool, L3 5DA, UK
[4]National Centre for Earth Observation, Leicester, LE4 5SP, UK
[*]Corresponding author: Dale Partridge, dapa@pml.ac.uk

**Abstract.** The spring bloom is an annual event in temperate regions of the North Atlantic Ocean in which the abundance of photosynthetic plankton increases dramatically. The timing and intensity of the spring bloom is dependent on underlying physical conditions that control ocean stratification and mixing. Although waves can be an important source of turbulent kinetic energy to the surface mixed layer, they have seldom been considered explicitly in studies of bloom formation. Here,

we investigate the role of surface waves in bloom formation using a combination of satellite observations and numerical models. Satellite observations show a positive correlation between wave activity and chlorophyll concentration in the Northwest European shelf (May-September). In the deeper Northeast Atlantic, increased wave activity correlates with lower chlorophyll during periods of high phytoplankton activity (March-May) and higher chlorophyll when activity is low (below $54°$N, July-September). We use a first-of-its-kind, km-scale, two-way coupled model system to investigate both the relationship between

wave mixing and bloom formation, and the sensitivity of model results to the method by which wave mixing is parameterised. In deep regions, during the spring bloom, a wave mixing event is likely to mix surface chlorophyll to deeper layers, away from light. In contrast, when and where phytoplankton activity is low in deep regions, wave mixing can entrain nutrients, fueling the growth of nutrient starved phytoplankton near the surface. In June to October, in shallow but weakly stratified regions of the shelf, surface chlorophyll tends to be elevated following a wave mixing event, which can bring to the surface

both phytoplankton from deeper layers and nutrients. When contrasted with ocean only runs, the two way-coupled ocean-wave model tends to produce greater vertical mixing and a delay in bloom onset. These results indicate bloom dynamics are sensitive to the way in which waves are modelled, and that the role of waves in bloom formation should be considered in future studies.

## 1 Introduction

The spring bloom is a defining feature of annual marine primary production in temperate waters of the Northeast Atlantic and

adjoining shelf seas. It is characterised by a rapid increase in the abundance of phytoplankton - photosynthetic microorganisms that are the ocean's dominant primary producers. Following winter, the spring bloom provides the first major influx of food



for the rest of the marine food web, and many organisms have adapted their development to take advantage of the increased food supply (Cushing, 1990; Ji et al., 2010; Cyr et al., 2024). The spring bloom is tightly linked to changes in the physical environment at the end of winter; a link first identified by G. A. Riley in the early 1940s (Riley, 1942). However, the exact

trigger for the spring bloom is still debated. Sverdrup proposed the Critical Depth Hypothesis, which states that the spring bloom occurs when the thermocline rises above a critical depth (Sverdrup, 1953). Sverdrup's critical depth is the maximum depth a phytoplankton cell can be mixed down to while, on average, receiving enough light to offset losses associated with respiration and other processes. When the thermocline is above the critical depth phytoplankton cells are trapped in well-lit, nutrient-replete waters near the surface where conditions are favourable for growth and reproduction.

Another proposed explanation for the timing of bloom formation is the Critical Mixing Hypothesis, which states that a bloom can form in a mixed water column if the rate of mixing is low enough for phytoplankton near the surface to achieve net positive growth (Huisman et al., 1999). This mechanism was explored by Taylor and Ferrari (2011) using high resolution, three-dimensional ocean models, with a reduction in net cooling at the end of winter identified as a key trigger. Biological

controls on bloom formation have also been proposed, including the Dilution-Recoupling hypothesis, which focuses on the balance between phytoplankton growth and grazing pressure through winter and into spring (Behrenfeld, 2010). Although the exact mechanism controlling bloom formation is still debated, a recent study using autonomous underwater gliders lent most support to Sverdrup's Critical Depth Hypothesis based on the depth of active mixing (Rumyantseva et al., 2019).

In tidally active areas on the continental shelf, stratification and bloom formation are also dependent on the water depth and the degree of mixing in the bottom boundary layer. Shallow regions of the Northwest European Shelf (NWES), including areas in the English Channel and the southern North Sea, remain permanently mixed year round (van Leeuwen et al., 2015). Other areas may be intermittently stratified, depending on prevailing tidal and atmospheric conditions. The bloom itself can be interrupted by an increase in bottom mixing associated with the spring-neap tidal cycle, or the passage of a storm, leading to

an apparent double bloom (Sharples et al., 2006).

Once established in deeper, seasonally stratified waters, stratification can be difficult to break down, requiring extreme atmospheric conditions, such as those associated with the passage of a hurricane or typhoon, to fully mix the water column (Shi and Wang, 2007; Babin et al., 2004; Sharples et al., 2001). Such events can trigger a new bloom in their wake, fueled by a fresh

influx of nutrients that have been mixed up from below the nutricline. In the absence of an extreme mixing event, the bloom will often peak before reducing in intensity as the supply of nutrients is exhausted and top-down pressures from grazing and viral lysis take hold (Simpson and Sharples, 2012). Near to the nutricline, given sufficient light, it is common for a sub-surface chlorophyll maximum to form in the summer. As stratification begins to break down at the end of winter, an autumnal bloom can sometimes be observed.




While the spring bloom is an annual event, its timing, duration and intensity exhibit significant inter-annual variability. Using models driven at the surface by winds and changes in the net heat flux, Waniek (2003) showed the spring-time shallowing of the mixed layer can be interrupted by mixing events caused by weather systems whose frequency and intensity vary from year to year, leading to interannual variability in bloom dynamics. This is a result that is embedded in contemporary coupled hydrodynamic-biogeochemical models, which are typically forced at the surface by a set of common variables, including wind, surface pressure, surface temperature, net shortwave radiation and fresh water fluxes.

One aspect that isn't typically considered explicitly in studies of bloom dynamics is the impact of surface waves on turbulent mixing. Traditionally, models for different components of the Earth system - the atmosphere, land, ocean and ocean waves - have been run independently. However, with improvements in computing power, studies using two-way coupled model configurations are becoming increasingly common (Berthou et al., 2025b). Wave breaking induces turbulent mixing which helps to stir the ocean's surface, with a direct feedback on atmosphere-ocean exchange. Recent studies have shown benefits of coupling a wave model to an ocean model in terms of marine forecasting of extreme wave heights and surges, ocean mixing (Lewis et al., 2019); and surface currents (Bruciaferri et al., 2021). Replacing the implicit calculation of wave-induced mixing in the ocean by explicit coupling to a wave model induces enhanced vertical mixing and better represents Stokes drift (Bruciaferri et al., 2021). This leads to a deeper summer mixed layer, resulting in a relative cooling of surface and upper ocean temperatures during periods of stratification (Lewis et al., 2019). The impacts of wave coupling on the ocean are generally weaker in shallow on-shelf areas, which is likely a result of them being mixed by tidal processes throughout the year (Lewis et al., 2019).

Liu et al. (2025) observed a delay in the spring phytoplankton bloom in the South China Sea when a parameterisation of wave induced mixing was added to a coupled hydrodynamic-biogeochemical model, suggesting a sensitivity to wave induced mixing. In this study, we investigate the sensitivity of bloom timing and ecosystem processes to wave coupling across the NWES and the Northeast (NE) Atlantic. This region is located at the end of the northern and central branches of the North Atlantic storm track (Woollings et al., 2010), and is therefore subject to high wave activity in the winter, and episodic wave activity in summer. The NWES is also an extremely productive region, with a pronounced spring bloom evident over large areas of the shelf (Simpson and Sharples, 2012). Specifically, we address the following questions: (i) What is the relationship between wave activity and phytoplankton bloom phenology? (ii) How sensitive are model results to the implicit and explicit representation of wave mixing effects?

For the study, we use the UK Met Office's operational ocean-wave coupled model system, which has been extended by coupling it to the European Regional Seas Ecosystem Model (ERSEM) (Butenschön et al., 2016). The new system allows two-way feedback between each model component to be represented, providing a more realistic representation of surface mixing and ecosystem responses. The study represents the first time a coupled ocean-wave-biogeochemistry model at km-scale resolution has been used to simulate ecosystem processes across the NWES and NE Atlantic. The modelling work is complemented by





90 an analysis of both satellite and in situ observations.

The plan for the paper is as follows. In section 2, we describe the study area and quantify relationships between satellite ocean colour and wave reanalysis data for recent decades. We then focus on the year 2018, when three named storms passed over the UK between the months of June and September, and examine the temporal evolution of phytoplankton abundance 95 over the course of the year. In section 3, we describe the modelling tools which we use to simulate the year 2018, showing the results of the modelling study in section 5. In section 5, we use the models and observations to compare bloom phenology and ecosystem responses to summer wave activity, and investigate the impacts that wave coupling has on ocean physics and biogeochemistry. Finally, in section 6, we bring the modelling and observation results together to propose mechanisms for the phytoplankton activity response to wave activity shown in section 3.

100

## 2 The Northwest European Continental Shelf and North East Atlantic

### 2.1 Physical and hydrodynamic characteristics

The NWES is a broad temperate continental shelf on the eastern side of the North Atlantic Ocean (Fig. 1). The region is demarcated by the shelf break, which extends northwestward through the Bay of Biscay in the south, along the edge of the Americon 105 shelf and up and around the west coasts of Ireland and Scotland, then northeastward toward the west coast of Norway. Topographic steering drives the northward flow of water parallel to the shelf slope, with more limited cross-slope transport driven by surface winds and meandering eddies. Off shelf, in the north of the region, the North Atlantic Current flows northeastward between Iceland and Scotland. Water enters the North Sea in the north through central and western regions of the northern North Sea, and in the south through the English Channel. The dominant outflow from the North Sea is via the Norwegian 110 Coastal Current.

On shelf, dynamics are controlled by seasonal changes in solar irradiance and heating, atmospheric fluxes and wind forcing, tides, river inputs and exchanges with the open ocean. Large areas of the shelf are seasonally stratified, including in the Celtic Sea, western reaches of the English Channel and the central North Sea. Shallower areas on the shelf may be intermittently 115 stratified or permanently mixed, with tidal mixing fronts separating stratified and well mixed areas. Nutrient concentrations are influenced by exchanges with the open ocean, atmospheric deposition and, particularly in coastal areas, river discharge. Bottom-up controls on the growth of phototrophic plankton include the availability of light and nutrients; and temperature. The spring bloom typically occurs from late March into April (Racault et al., 2012), and tends to start in the south of the domain before spreading northward. In coastal waters with high sediment loads, the growth of surface plankton can be light limited. In 120 the summer, under stratified conditions, surface phytoplankton become nutrient limited.





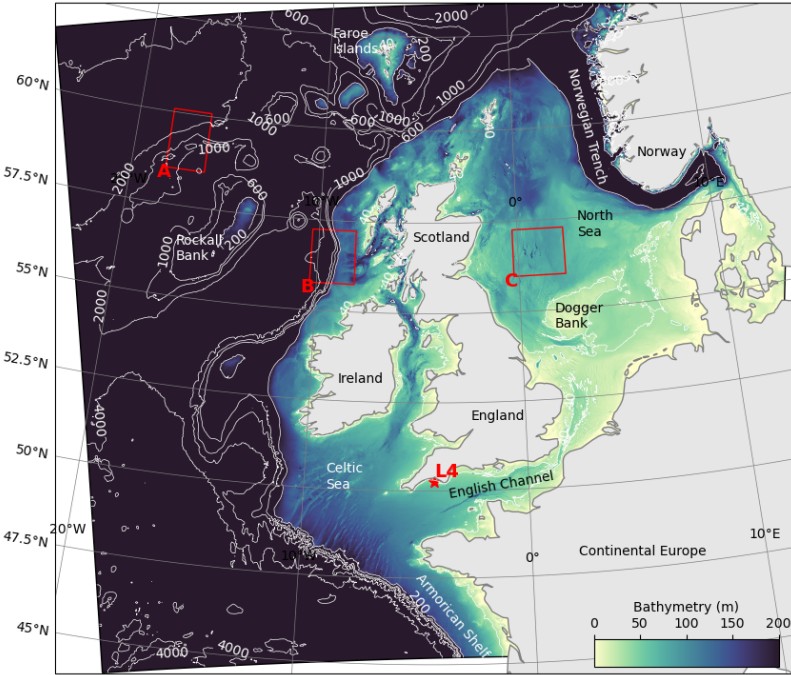

**Figure 1.** AMM15 model domain extent with bathymetry contours (m). Red boxes highlight representative analysis zones for off-shelf (A), shelf break (B) and on-shelf (C) areas. The L4 monitoring station is shown with a star.

## 2.2 Monthly wave energy relationship to observed chlorophyll concentration in climatology

To explore the potential impact of waves on bloom dynamics in the NE Atlantic and on the NWES, we examine satellite ocean colour and wave reanalysis data. We used chlorophyll-a from the Ocean Colour Climate Change Initiative (OC-CCI) dataset (Sathyendranath et al., 2019, 2023) and wave energy from the Met Office regional wave hindcast from WAVEWATCH

III (Saulter, 2024) to calculate interannual correlations between bimonthly-mean chlorophyll concentrations and wave energy from 1998 to 2020. To achieve this, the temporal mean was calculated across pairs of months for each year from March to November, when the majority of primary production occurs. Wave energy and chlorophyll data were then normalised at each spatial pixel and the Pearson correlation coefficient calculated between them, with significance tested using a two-sided p-value for the correlation being different to zero (Virtanen et al., 2020).


The results show a strong negative correlation between wave energy and chlorophyll concentration off shelf and near the shelf break in March-May, and a strong positive correlation on-shelf from June to October (Figure 2). This suggests that stronger wave activity in the open ocean when phytoplankton are blooming in March-May leads to a reduction in surface chlorophyll, whilst the reverse is true on-shelf later in the year. Off shelf, the response to enhanced wave activity appears positive in the

southwest of the domain from June to September. In the northwest of the domain, the open-ocean response is confined to close





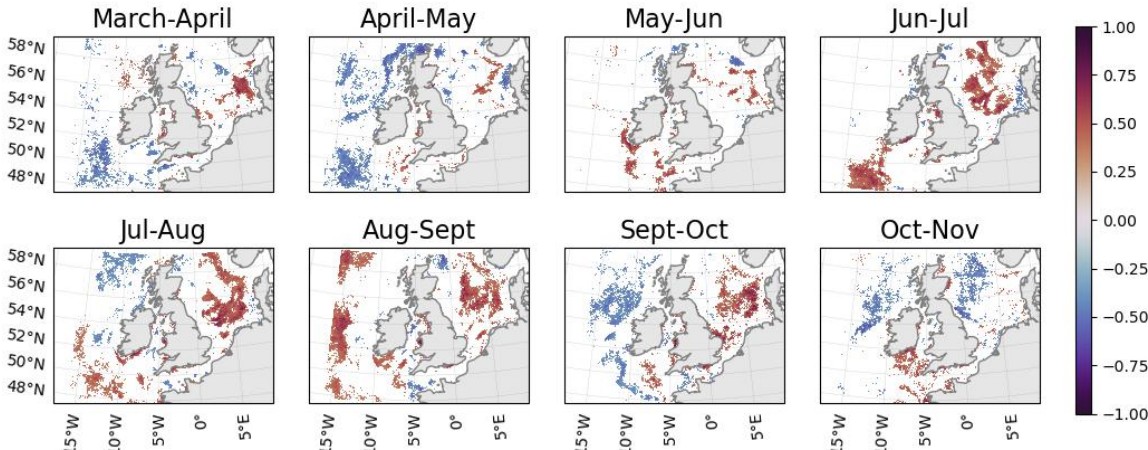

**Figure 2.** Correlation between mean wave energy and mean chlorophyll concentration from 1998 to 2020 for different pairs of months. Only regions where the correlation coefficient is significant (p<0.1) are coloured. Wave data is from a WAVEWATCH III reanalysis (Saulter, 2024) and chlorophyll satellite data is from OC-CCI (Sathyendranath et al., 2019, 2023).

to the shelf break, and is generally negative, except in August-September.

## 2.3 Wave activity and chlorophyll concentration in 2018

The year 2018 has been selected for this study as there was a cooler ocean surface in March-April, when the spring bloom
typically occurs, and enhanced wave activity compared to climatological conditions (Fig 3). Later in the year, two marine heatwaves (MHW) developed. The first was in late May to mid-June and the second in July. The first was terminated by storm Hector, which passed over the UK between 13th and 14th June 2018. Storm Hector brought heavy rain and winds to the North of the UK and Ireland. In Northern Ireland, maximum gust speeds in excess of $30\,\mathrm{m\,s^{-1}}$ were recorded. The second heatwave was terminated by an unnamed storm which passed over the UK on 29th July. Storm Ali and Bronagh passed over the UK in
succession between 18th and 21st September.

We used the approach of Jardine et al. (2022) to determine the timing of bloom onset. In this approach, bloom onset is defined as the start of exponential growth in the concentration of chlorophyll, which is used as a proxy for phytoplankton biomass. This is defined to be the time when the change in the concentration of chlorophyll exceeds $0.15\,\mathrm{mg\,m^{-2}\,d^{-1}}$ consistently for
5 consecutive days, and is referred to as the date of exponential onset. This approach performs best with full temporal data coverage, ideally suited to evaluating model data. For satellite data with gaps in coverage the algorithm occasionally fails to identify the bloom, however for consistency we use the same approach for satellite and model data.



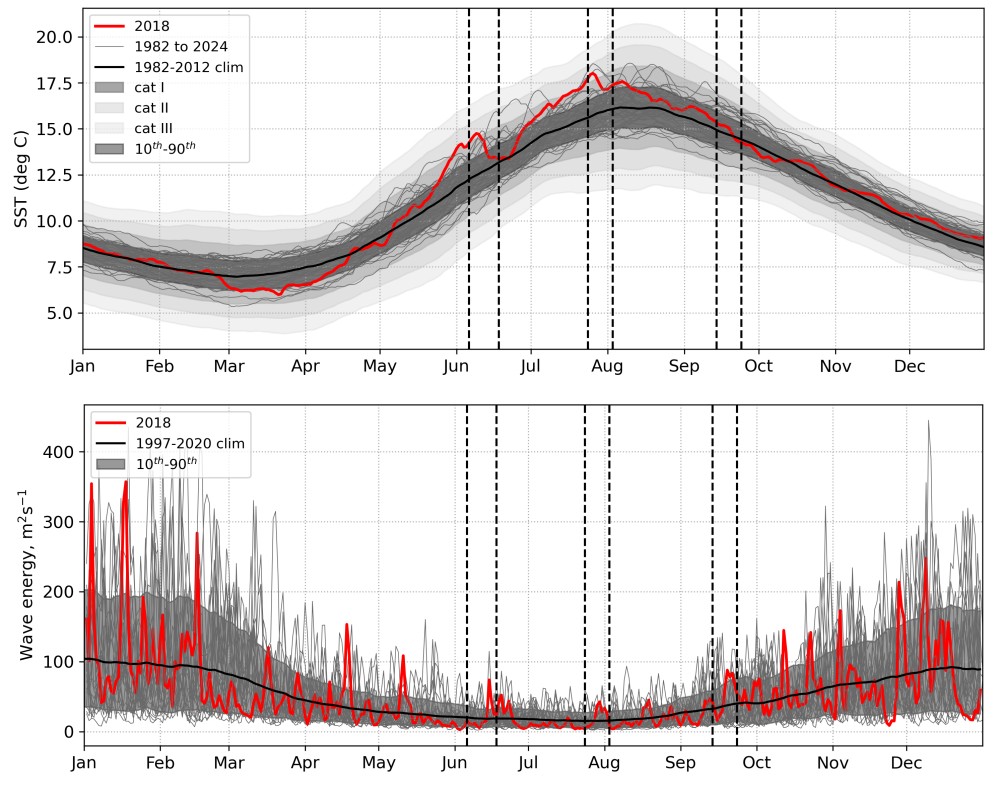

**Figure 3.** Top - Operational Sea Surface Temperature and Ice Analysis (OSTIA) SSTs for 1982–2022, mean climatology for the time period used to define marine heatwave thresholds (1982–2012) in bold, 10th–90th centiles (anomalies smoothed with 31-day moving average), Shading: Category I, Category II, Category III marine heatwaves using Hobday et al. (2018) averaged over the NWS. Bottom - Wave energy from Met Office wave regional reanalysis (Saulter, 2024) for 1997–2020. Yearly data shown with grey lines, mean climatology from 1997–2020 in black, 2018 in red. Grey shading between 10th–90th centiles. Mean and centiles smoothed with 31-day moving average. Wave energy averaged over the NWS (NWS=blue+purple regions in Fig. 1). Vertical dashed lines indicate three key 2018 storm periods.

In 2018, phytoplankton bloom onset happened around mid-April in the south of the domain and drifted into May in the North and West, with large spatial variability (Figure 4). On the shelf, bloom onset also generally occurred in April, although the bloom began later on the northern shelf break and parts of the Channel and Celtic Sea.

Late summer storms can either suppress or enhance chlorophyll, depending on timing and area (Fig. 5). In June, as a result of storm Hector, the concentration of chlorophyll was reduced in northern off-shelf areas, which were in bloom state, whereas chlorophyll increased on the western part of the shelf in regions which were past their bloom state. Particularly strong increases are noticeable in the Celtic Sea, Irish Sea, and Southern North Sea. Following the July storm, chlorophyll concentration was low off shelf and the storm had a minimal impact. In contrast, the concentration increased in coastal areas of the shelf. In



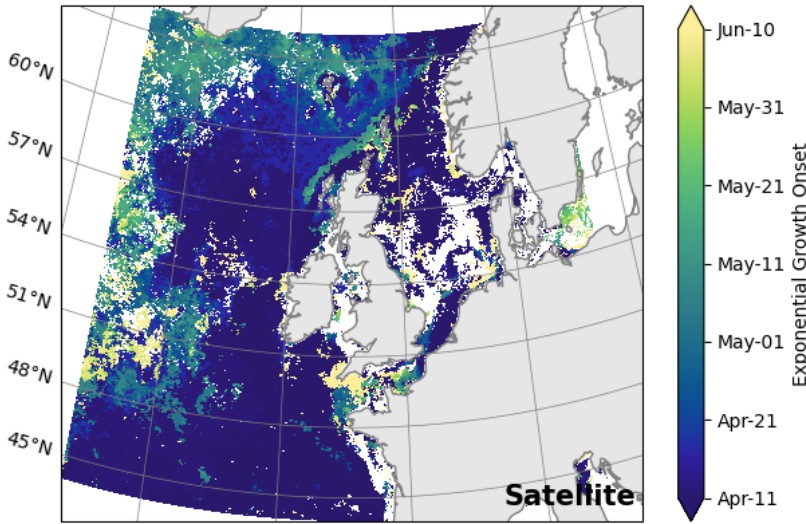

**Figure 4.** Date of onset of exponential phytoplankton growth in spring 2018 satellite data from OC-CCI (Sathyendranath et al., 2019, 2023). Bloom onset calculated using the method from Jardine et al. (2022), where bloom onset is defined to occur when phytoplankton biomass increases by $0.15\,\mathrm{mg\,m^{-2}\,d^{-1}}$ consistently for 5 consecutive days.

September, northern areas tended to see a decrease in chlorophyll from storms Ali and Bronagh, whilst southern and on-shelf areas saw a strong increase.


The observations confirm 2018 to be an interesting year to study the relationship between wave activity and the concentration of chlorophyll and suggest that waves played a role in controlling chlorophyll concentrations in that year. In the remainder of the article, we focus on how intense periods of wave activity affected the concentration of chlorophyll, and how explicitly coupling an ocean model to a wave model impacts model results.


## 3 Model Framework

The impact of ocean-wave coupling on biogeochemistry is assessed through a twin experiment. Experiments are conducted with (WAV) and without (OCN) the coupled wave model in order to examine the biogeochemical response, as shown in Scheme 1.


**Figure 5.** Mean chlorophyll from satellite in the five days before the storm (left), wave activity on the day of the storm from the Met Office wave regional reanalysis (Saulter, 2024) (middle) and difference in the mean satellite chlorophyll five days after and before (right) storm Hector (top), July storm (middle) and storms Ali and Bronagh (bottom). The start of the five days before and the end of the five days after the storms are highlighted in Figure 3 as vertical bars.



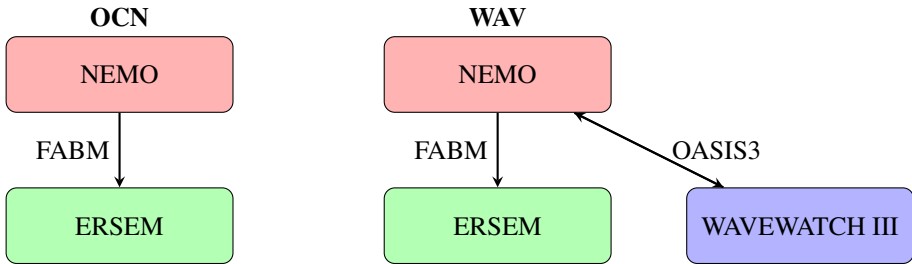

Scheme 1: Twin experiment configurations

## 3.1 Hydrodynamic Model

Both simulations have been performed using the Nucleus for European Modeling of the Ocean (NEMO) model (Madec et al., 1998), adapted for simulating shelf sea dynamics (O'Dea et al., 2012). For this study, we use NEMO v4.04. The model domain is the high-resolution $1.5\,\mathrm{km}$ Atlantic Margin Model (AMM15) configuration (Fig. 1), which is used for operational forecasting by the UK MetOffice (Tonani et al., 2019). In contrast to the more commonly used $7\,\mathrm{km}$ configuration for the same region, the high resolution domain enables smaller scale features such as eddies and internal tides to be better represented (Graham et al., 2018a, b).

Initial fields for the simulations were provided by a previous ocean-wave experiment (Lewis et al., 2019), with lateral boundary forcings from global ocean models. Meteorological forcing is provided at 3-hourly intervals from the UK Met Office global Unified Model analysis, at a 25km resolution. There is no live coupling from sea-surface back to the atmosphere.

For the OCN simulation, the vertical eddy viscosity is computed using the two-equation GLS turbulent closure model. In the absence of a two-way coupled wave model, the impact of waves on the ocean is parameterised in the following way (see Reffray et al. (2015) for the details about the NEMO implementation):

  – surface enhanced mixing due to wave-breaking according to Craig and Banner (1994) scheme;

  – sea surface roughness as a function of the significant wave height, approximated based on wind speed, as proposed by Rascle et al. (2008).

Additionally, it is assumed that the water-side momentum flux (i.e., the wind stress $\tau_{atm}$) is completely transferred into the ocean.

## 3.2 Wave Model

The wave component of the NWS prediction system is AMM15-wave, a regional implementation of the WAVEWATCH III spectral wave model version 4.18 (Tolman, 2014) as detailed in Saulter et al. (2017). The domain of AMM15-wave covers





the same area as the AMM15 NEMO model but uses a Spherical Multiple Cell discretization scheme (Li, 2012) configured to have a variable horizontal resolution ranging from $3\,\mathrm{km}$ across much of the domain down to $1.5\,\mathrm{km}$ near the coast or where the average depth is shallower than $40\,\mathrm{m}$. Wave growth and dissipation terms are parameterized following Filipot et al. (2010) while nonlinear wave-wave interactions use the Discrete Interaction Approximation (DIA) package according to Hasselmann
et al. (1985).

### 3.3  Wave-Ocean coupled model

Two-way coupling utilises the OASIS3-MCT coupler (Valcke et al., 2015). NEMO shares water levels and currents with the spectral wave model. In turn, WAVEWATCH III injects Stokes drift, significant wave height and water-side momentum flux
into the hydrodynamic model. The coupling frequency is hourly.

In this system, the momentum budget equation solved by the ocean model is modified to include three wave feedbacks as described in Bruciaferri et al. (2021). The first is the Coriolis-Stokes force (CSF), an interaction between the Stokes drift in the direction of propagation of waves and the Coriolis force. The second is a wave modified water-side momentum flux: the
wavefield is not always in equilibrium with the local wind as assumed in the standalone ocean model (especially in coastal areas like the NWS). The waves are either growing, with a net influx of momentum into the wavefield, or decaying, with intensified wave-breaking and a net outflux of momentum from waves into the ocean (e.g., Komen et al. (1994)). Lastly, a sea-state dependent sea surface roughness (calculated from the significant wave height), is taken into account in the calculation of the vertical eddy viscosity, as explained in Section 3.1. Note the wave-breaking term is still calculated by the ocean only system,
but will be modulated by the change in water-side momentum described above.

### 3.4  Biogeochemical Model

For this study, we use the European Regional Sea Ecosystem Model (ERSEM) for simulating ocean biogeochemistry and phytoplankton bloom dynamics (Butenschön et al., 2016). ERSEM has been coupled to NEMO using the Framework for
Aquatic Biogeochemical Models (FABM) (Bruggeman and Bolding, 2014). FABM facilitates passing data between biogeochemical and hydrodynamic models. Here, the coupling is one way, and the biogeochemical model does not feedback onto ocean physics. Whilst ERSEM is not directly coupled to the wave model, biogeochemical processes are influenced by changes in ocean currents and mixing that are caused by the wave model.

**Biogeochemistry Initial & Boundary Conditions**

To the authors' knowledge, at the time of simulation this is the first instance of ERSEM being used with the NEMO AMM15 domain. Initial values are interpolated from a reanalysis run performed on the coarser AMM7 domain for November 2016. Since the north-west corner of the AMM15 region extends beyond the bounds of the AMM7 domain, extrapolation is performed




using a simple nearest neighbour algorithm. Before performing the 2018 experiment, a fourteen month ocean-biogeochemistry

only simulation starting on 1st November 2016 is used to 'spin-up' the biogeochemical initial conditions.

A constant supply of non-depleting nutrients at the boundary can lead to spurious phytoplankton blooms and other unrealistic responses. This is mitigated by enforcing a low, constant value at the boundary for most of the previously unconstrained tracer fields (Polton et al., 2023). Biogeochemical surface boundary conditions include nitrogen deposition from the atmosphere and

light attenuation due to detritus and yellow matter (the Gelbstoff absorption coefficient). Nitrogen deposition data is available at monthly resolution using models run by the European Monitoring and Evaluation Programme (EMEP) (Simpson et al., 2012), which are then converted into fluxes for both oxidised and reduced components. The Gelbstoff absorption coefficient is produced using data from OC-CCI (Sathyendranath et al., 2019), for a multitude of wavelengths that are integrated into a single broadband field.


River input data is an updated version of the files used in the CMEMS northwest European shelf reanalysis from Lenhart et al. (2010). A climatology from the Global River Discharge Database (Vörösmarty et al., 2000) and data from the Centre for Ecology and Hydrology (Young and Holt, 2007) have been used to provide time varying daily river discharge, nutrient loads (nitrate, ammonia, phosphate and silicate), total alkalinity, dissolved oxygen and dissolved inorganic carbon.


## 4   Results

### 4.1   Validation

In Lewis et al. (2019); Graham et al. (2018a, b), a detailed validation of the physical ocean and wave models has been performed, demonstrating good performance of standalone models and improvements to significant wave height and total water

level extremes when coupling ocean and waves (Lewis et al., 2019). For biogeochemistry, while there have been several modeling studies using ERSEM on the coarser AMM7 domain (e.g. Jardine et al., 2022; Powley et al., 2024), this experiment is the first simulation of ERSEM biogeochemistry on the $1.5\,\mathrm{km}$ regional domain.

Profiles of temperature across depth from both the OCN and WAV models were compared to profiles of temperature ex-

tracted from the EN4 Met Office Hadley Centre Observation Dataset (Good et al., 2013) for March to June (Fig. 6) in off-shelf areas deeper than $200\,\mathrm{m}$. Both models show a persistent warm bias off-shelf, which is the result of using surface forcing that itself has a warm bias as well as the model drifting before the start of our 2018 simulations. The coupling of waves results in a marginal improvement during spring when the water column is well mixed, but as it stratifies in June the impact of waves increases leading to a reduction in the overall bias.




**Figure 6.** Potential temperature profiles within the off-shelf region, averaged over all observed profiles in the EN4 database that are located within 1.5 km of a model grid cell for March, April, May and June (blue), along with the corresponding simulated profiles at the same point and time for the OCN (orange) and WAV (green) simulations.





| Variable | Observed Mean | Model Mean | Bias | Spatial Correlation | Temporal Correlation |
|---|---|---|---|---|---|
| Temperature ($^{\circ}C$) | 11.69 | 12.54 (12.37) | 0.85 (0.68) | 96.7% (97.2%) | 98.0% (98.3%) |
| chlorophyll ($mg/m^3$) | 0.83 | 1.08 (1.08) | 0.25 (0.25) | -0.14% (-0.13%) | 48.9% (47.2%) |

**Table 1.** Summary statistics for key variables at the surface matched against satellite observations for temperature (OSTIA) and chlorophyll (OC-CCI). Results from the simulation with wave coupling are provided in parentheses.

The warm bias for temperature is also present when the ocean-only model is compared to satellite surface fields from the Operational Sea Surface Temperature and Ice Analysis (OSTIA) (Fig. 7), with a persistent positive bias across most of the domain. The bias is largely similar between the OCN and WAV simulations, with both models able to capture spatial and temporal patterns in temperature (Table 1).


For chlorophyll, there is also a positive bias across the off shelf area of the domain compared with OC-CCI surface satellite measurements for 2018 (Fig. 7), suggesting an overprediction of phytoplankton activity. On-shelf and coastal areas have a negative bias, with the largest bias in areas close to river mouths. In part, the bias is a result of using climatological river input data, which prevents the model from capturing inter-annual variability in river inputs and their impact on ecosystem dynamics. Table 275 1 suggests that the model can reasonably capture the spatial variability. The aforementioned difficulties around the coast and in regions of fresh water influence, mean the model struggles to capture the same temporal patterns seen in the observations. However it is difficult to draw many conclusions for chlorophyll as statistics for a single year will be dominated by the timing and strength of the spring bloom.

For in situ observations of biogeochemical variables, we use data from the long-term monitoring site Station L4 (McEvoy et al., 2023), which is located on the western edge of the English channel (Fig. 1). The area is influenced by freshwater inputs from the river Tamar. While the use of climatological river forcing data means the model will fail to reproduce inter-annual variability in river inputs, the data provides an opportunity to investigate the relative impact of wave coupling. Fig. 8 shows a comparison of the two simulations with observations for surface temperature, salinity and chlorophyll-*a* from Station L4. Also 285 plotted are Net Primary Production (NPP) from both model runs and Significant Wave Height from the WAV simulation. Temperature is consistent between the models and correlates well to the observations, with the warm bias evident over the winter months and the effect of the marine heatwaves visible in the observations (Fig. 8a). In winter months, the WAV model is more saline at the surface (Fig. 8b), indicating greater mixing of buoyant, freshwater inputs from the river with more saline waters. These periods typically correspond to times of higher wave activity (Fig. 8e). The inclusion of waves has a minor impact on 290 the spring bloom (Fig. 8c), with slower initial growth and a slightly delayed peak. Here, the model predicts the bloom a few weeks before the observations suggest it is occurring, with a late bloom seen in the area in the satellite data (Fig. 4). The trends identified in chlorophyll are also evident when examining net primary production (Fig. 8d). The same pattern of a short period of lower values in the WAV run followed by a longer period of higher values is also evident in September at the time of the




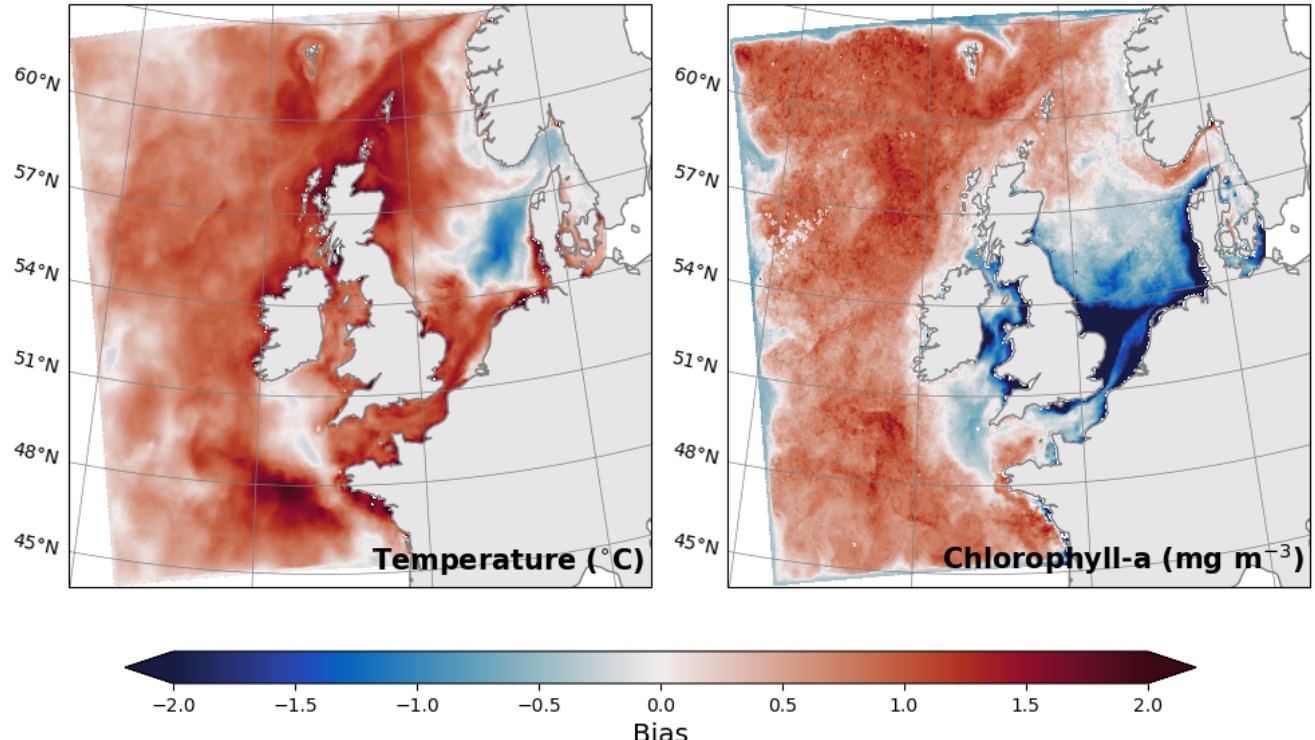

**Figure 7.** Annual mean bias at the surface between the ocean-only model and (left) satellite temperature from OSTIA, or (right) chlorophyll from OC-CCI.

autumn bloom.


## 4.2 Bloom phenology

Coupling with waves delays the bloom onset across most of the model domain (Fig. 9). On-shelf, the response to wave coupling is minimal as the shallow waters are already well mixed by tidal processes, whereas off-shelf bloom onset is typically 1-2 weeks later with larger delays in the south. In addition to delaying bloom onset, the yearly total of chlorophyll averaged

over the euphotic depth is up to $20\,\%$ higher in the high-productivity region along the northern reaches of the shelf break and also in some deeper waters (Fig. 9).

Relative to the date of bloom onset derived from satellite observations of sea surface chlorophyll (Fig. 4), the date of bloom onset in the model runs is around 1 month later. Delays in bloom onset relative to observations have been observed in past

NEMO-ERSEM simulations (Skákala et al., 2020), and, in part, stem from the parameterization of photosynthesis, phytoplank-



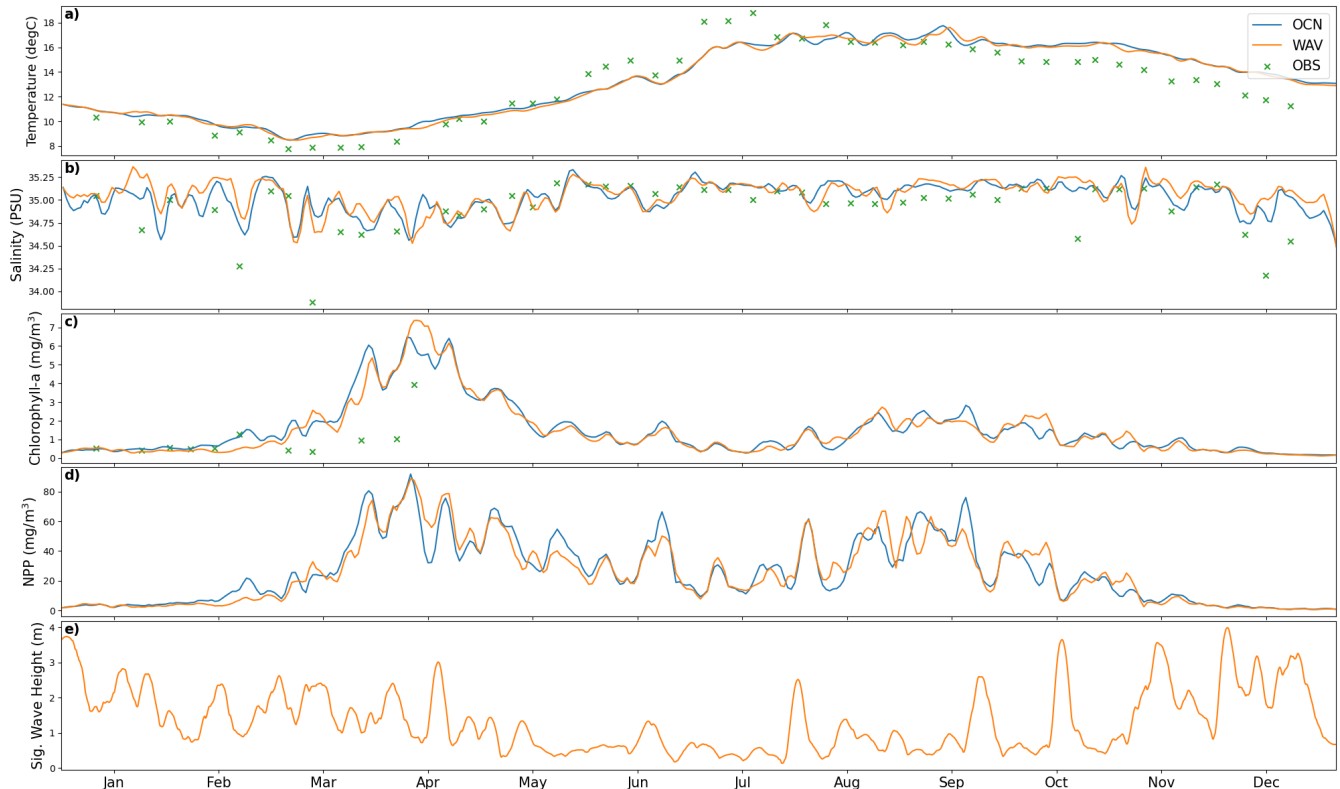

**Figure 8.** Surface comparison of modelled and observed biogeochemical variables at the L4 station (Fig. 1). Observations are from the Western Channel Observatory (McEvoy et al., 2023).

ton growth and grazing effects. The biological model is also influenced by biases in the physical model, including temperature, which directly impacts biological rates as well as the timing of stratification; and the ability to accurately represent optically active components, such as coloured dissolved organic matter (cDOM) and suspended particulate matter (SPM), which impact light attenuation. Nevertheless, spatial variability is captured by the model, such as pockets of very late bloom over the shelf;

and the later bloom onset along the shelf break north of Scotland and in the open ocean around 51 N. The south-north gradient is also evident in both the model runs and observations, although it is more pronounced in the model.

### 4.2.1 Response of phytoplankton to summer wave activity

Figure 10 shows the phytoplankton response to the three summer storms. The response can be contrasted with the observed

response, as inferred from satellite observations (Fig 5). It is necessary to analyse the response in the context of the later, simulated bloom onset date. In June, this results in chlorophyll concentrations that are generally higher in the model relative to observations. As a result, in absolute terms, Storm Hector has a larger impact on sea surface chlorophyll concentrations than it





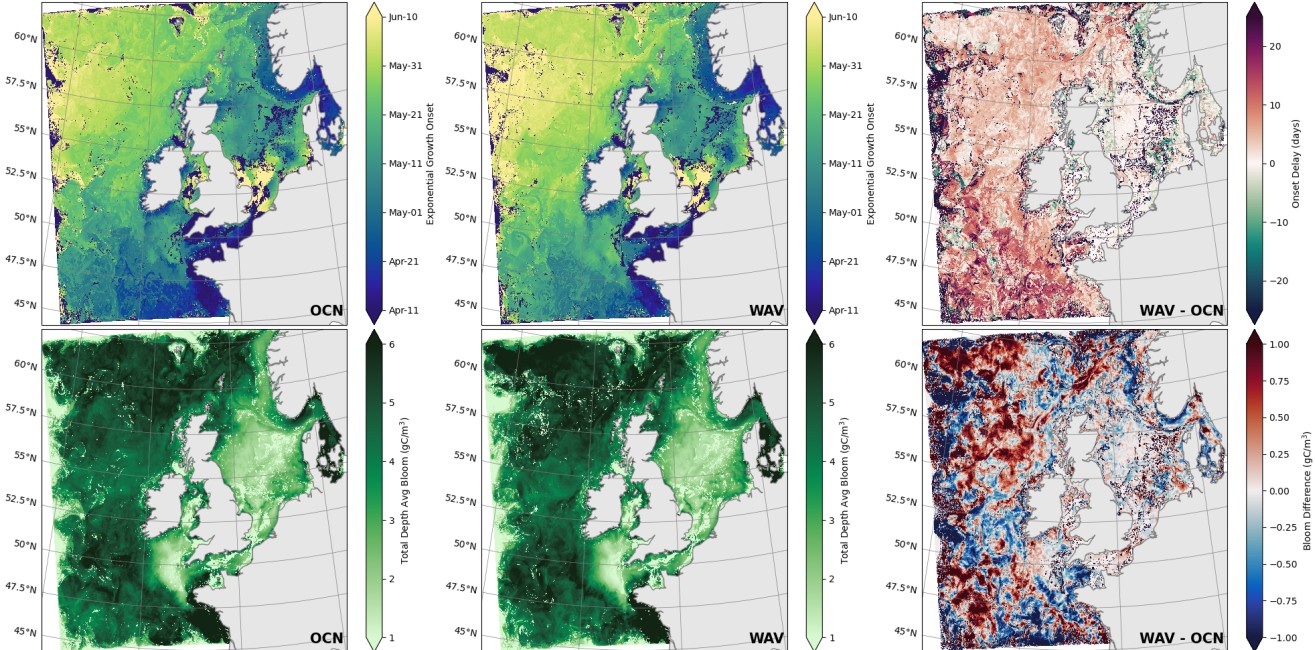

**Figure 9.** Date of onset of exponential growth (top) and total NPP over the year averaged to the euphotic depth (bottom) for OCN (left), WAV (middle) and the difference; WAV minus OCN (right).

does in the observations. The spatial pattern of the response to the storm is relatively similar between the model and the observations, with a suppression of phytoplankton activity off-shelf and increased activity on-shelf. For the July storm, the model

represents well the increase in the concentration of chlorophyll on the western and northern edges of the North Sea. However, there is little effect of the storm off-shelf in the observations, whereas the model shows an increase in the concentration of chlorophyll. For storms Ali and Bronagh, the north-south gradient in the response in the North Sea is well captured by the model, though it is not as intense.

Coupling with waves deteriorates the model response to the June storm due to the bloom occurring later, which results in a phytoplankton reduction rather than increase west of Ireland. Similarly, in September, the wave model reduces the areas where chlorophyll concentrations increase in the southwest of the domain, which is less in line with a widespread off-shelf increase in the observations.





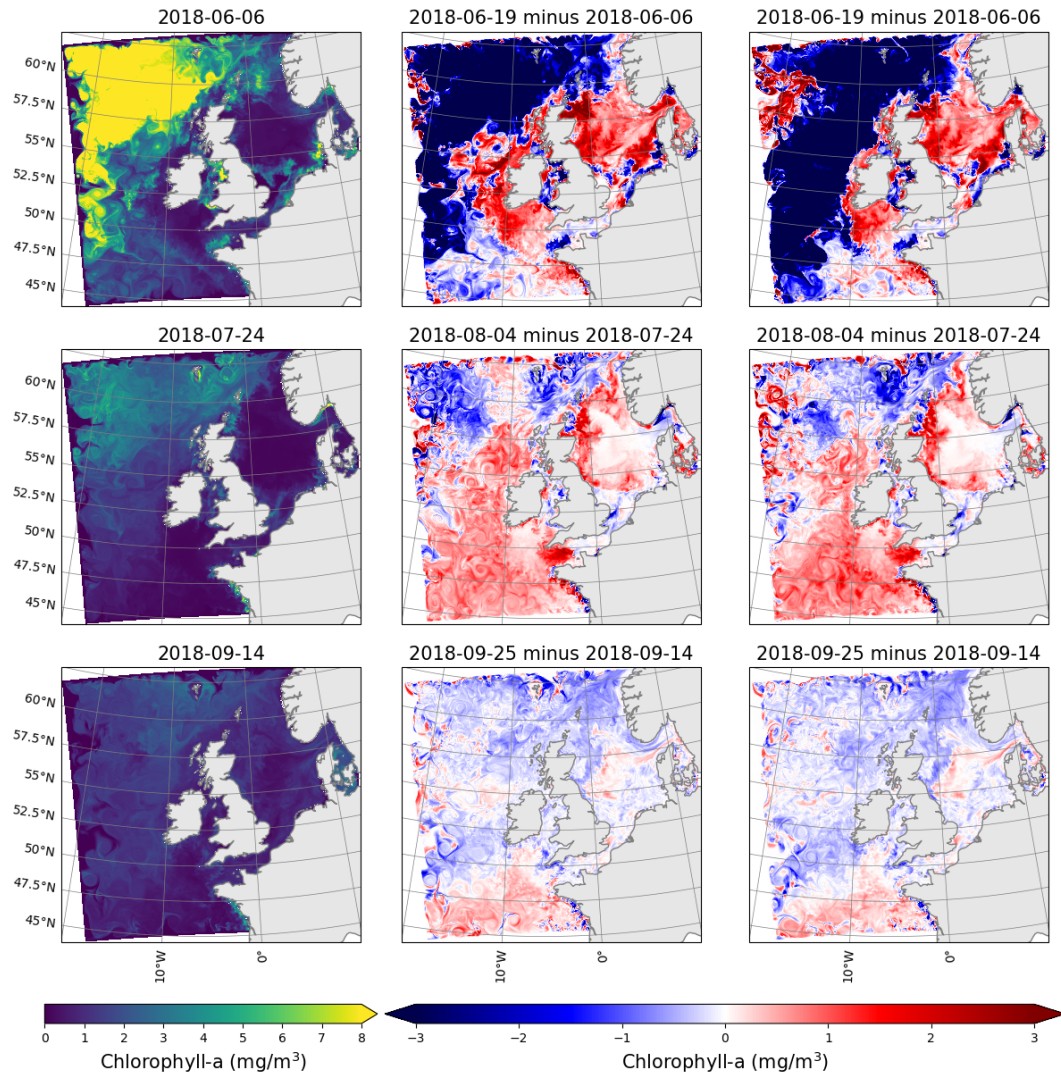

**Figure 10.** Mean chlorophyll in the five days before the storm from the ocean-biogeochemistry model (left), the difference between the mean of the five days after the storm and the mean of the five days before the storm from the ocean-biogeochemistry model (middle) and the ocean-wave-biogeochemistry model (right), for storm Hector (top), July storm (middle) and storms Ali and Bronagh (bottom). The start of the five days before and the end of the five days after the storms are highlighted in Figure 3 as vertical bars.





## 5 Wave Coupling Impacts

In this section, we carry out a more detailed analysis of the impact of wave coupling on biogeochemistry by assessing the differences between the two simulations. This allows us to isolate the impact of wave coupling, while accounting for biases in the models relative to observations. Chlorophyll fields are averaged over the euphotic zone, to the depth at which light is $1\%$ of its surface value, in order to capture sub-surface chlorophyll maximums.

### 5.1 Phytoplankton changes with wave coupling

Fig. 11 shows the simulated chlorophyll concentration throughout the year for a transect in the off-shelf region running N-S. In both the OCN and WAV simulations the bloom originates in the warmer, southern part of the domain with a strong signal around April followed by a period of elevated concentrations that eventually die out in January. In the northern part of the domain the initial bloom occurs approximately 2 weeks later, with the elevated concentration period ending by November. The bloom is interrupted by storm Hector mid-June and a secondary bloom is clear on the onshelf part of the transect ($48\,\mathrm{N}$ to $55\,\mathrm{N}$) at the start of July, which corresponds to the start of the marine heatwave (Fig 3), two weeks after storm Hector. This second peak is also present in observations, though 1 week earlier in the observations (not shown).

The differences between the two simulations are characterized by a mean delay of $3.2\,\mathrm{days}$ in the initial bloom when waves are coupled, as shown in Fig. 9. The WAV simulations additionally show increased chlorophyll concentration in the secondary late June peak, and then in August and September near the shelf break between $48\,\mathrm{N}$ and $55\,\mathrm{N}$.

By considering the on-shelf, shelf-break and off-shelf regions, as highlighted in Fig. 1, the temporal patterns for the different zones can be assessed. Fig. 12 shows the difference in mean net primary production (NPP) as a time-series. The shallow, sheltered on-shelf region is relatively less affected by the addition of waves into the system, whereas the shelf-break and off-shelf zones have a more pronounced difference. In both these regions there is a negative peak followed by a positive one, highlighting the offset nature of the blooms. In the shelf-break region, after the initial blooms for both models NPP levels are consistently higher throughout the summer when waves are included. In off-shelf areas, the offset is greater leading to a larger overall difference in primary production.

### 5.2 Physical changes with wave coupling

The top row in Fig 13 and Fig 14 show similar results to Lewis et al. (2019): when the water column in stratified, the ocean-wave coupled system is cooler in the mixed layer (and warmer below the mixed layer in May-June, before the first summer storm) due to increased vertical mixing when treating waves explicitly. Over the shelf region, this signal is also clear at the start of stratification episodes (May and July), but then reverses after strong mixing events, unlike in Lewis et al. (2019), where it remained for the whole season, as the stratification was not disrupted by storms in their chosen summer (2017). These mixing





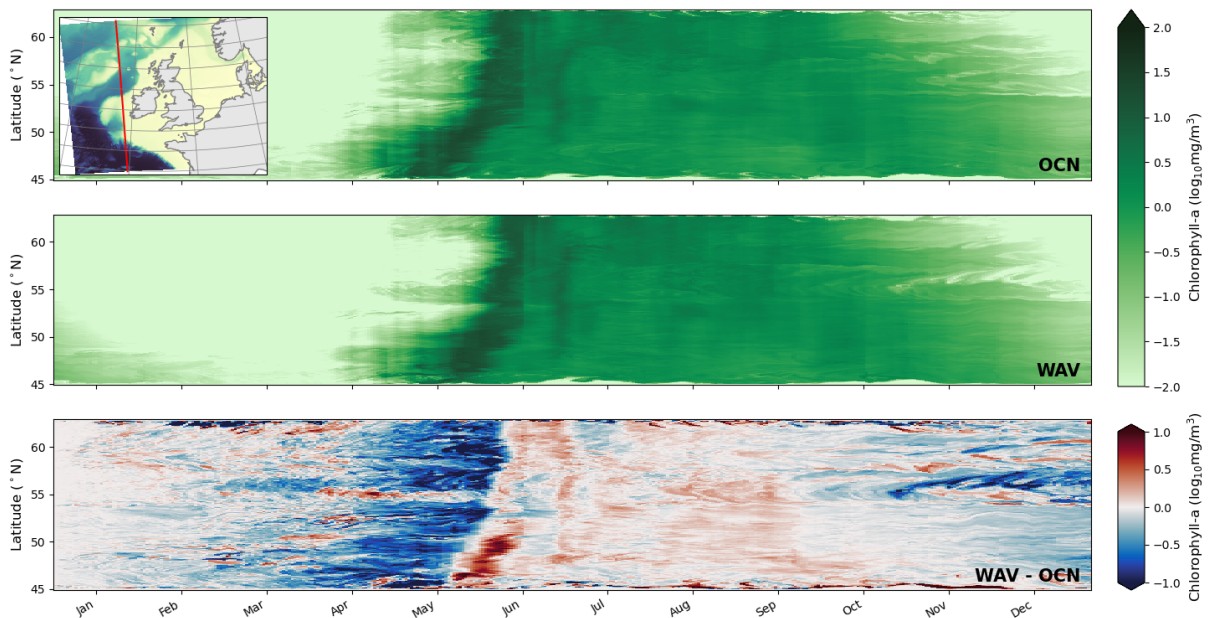

**Figure 11.** Chlorophyll (averaged over euphotic depth) time/latitude transect along the red line indicated in the top left hand inset for (top) OCN, (middle) WAV, (bottom) WAV-OCN

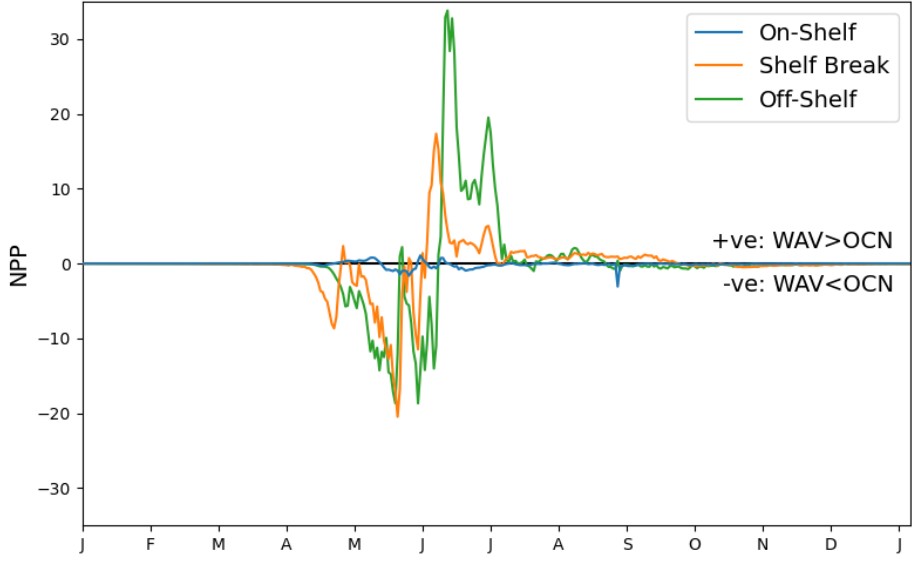

**Figure 12.** Mean difference in NPP between simulations for off-shelf, shelf-break and on-shelf regions as defined in Fig. 1





events (storm Hector (13-14 June), July 29 storm, storms Ali and Bronagh 18-21 September) are clearly seen through a sudden erosion of stratification in Fig 13. Vertical mixing is increased when coupled with waves (2nd row of Fig 14), and the coupled simulations are generally cooler from the surface down to 60 m to 100 m in the off-shelf and shelf-break regions after storms.
This persists even after stratification has re-formed (end June-July) due to a deeper mixed layer in the coupled runs.

The second row in Fig 13 and Fig 14 shows the main effect of coupling with waves off shelf and in the shelf-break is to increase turbulent kinetic energy (TKE) in the ocean, which is related to the change in surface roughness and water-side stress, now calculated by the wave model. On the shelf break, TKE penetrates deeper in the ocean during summer storms, potentially
because of the interaction with the bathymetry and breaking of internal waves. This explains the deeper cooling effect of wave coupling in this region.

On-shelf in the North Sea, the effect is more varied, with a decrease in TKE for the main mixing events during the stratification period (Fig 14, rightmost panels). This is likely due to the fact that wave age in the North Sea tend to be small as waves
there are generated by local wind sea, not remotely generated swells. Therefore, in this region, and in case of intense events like the ones responsible for deep mixing, the waves will tend to extract momentum from the atmosphere before it is passed to the ocean, and therefore reduce TKE (Gentile et al., 2021). This explains the warming with wave coupling after mixing events: the mixing in the ocean got less deep with wave coupling as waves extract energy from the atmosphere before it is passed to the ocean in this region.


## 5.3 Linking physical and biogeochemical changes from wave coupling

Nutrient differences with wave coupling are consistent with temperature changes once the stratification is established (Fig 13 and Fig 14, 3rd row): with decreased off-shelf and shelf break nitrate concentrations as it is consumed by phytoplankton. Extra TKE induced by the wave model brings additional nutrients from nitrate-rich deeper layers, reducing nutrient content in these
deeper layers. To the contrary, over the shelf where the impact of TKE is limited even during strong mixing events, nitrates are only slightly reduced in the WAV simulation, though the levels are very low anyway. Each storm imprint is also clearly seen in all profiles, increasing nutrient concentration near the surface, and this effect is stronger with wave coupling. On the shelf break, where TKE mixing affects deeper layers, phytoplankton is distributed more evenly throughout the column and the euphotic zone is deeper than 100 m. The additional mixing in this region results in more nutrients throughout 100 m depth,
and increased chlorophyll throughout the column (Fig 13 and Fig 14, 4th row). Off shelf, chlorophyll is only increased at depth under the euphotic zone due to increased mixing. Off-shelf, increased nutrient concentration near the surface may be compensated by decreased light availability with a deeper mixed layer.

At the onset of stratification (April-May south off-shelf and May-June in the North), the pattern in nitrate near the surface
is anticorrelated with the pattern of chlorlophyll-a content of Fig 13 - this pattern is controlled by the phytoplankton bloom,





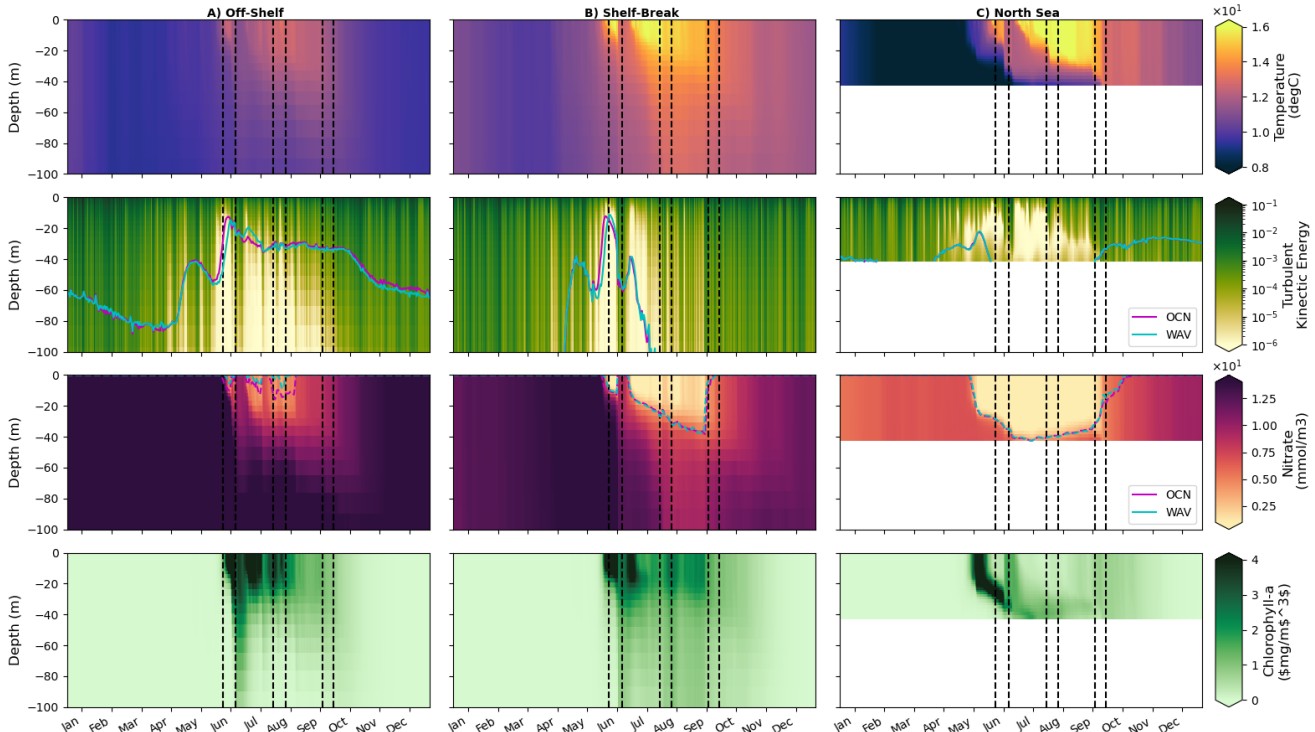

**Figure 13.** Hovmöller plots (depth/time) of WAV output for potential temperature, turbulent kinetic energy, nitrate, chlorophyll and light availability for (left) off-shelf, (middle) shelf-break and (right) North Sea regions defined in Fig. 1. Dashed lines indicate Magenta (OCN)/Cyan (WAV) time series show euphotic depth (1 % of surface light) in the second row and nutricline ($3\,\mu\mathrm{mol\,kg^{-1}}$ threshold) in the third.

which occurs later with wave coupling. After this, cholophyll-a content under the mixed layer is generally larger when coupled to the wave model, illustrative of a stronger export of phytoplankton at depth with stronger TKE.

The delay in phytoplankton bloom can be explained by the change in light availability for the phytoplankton at the infancy
stage of the bloom. At the onset of the stratification period, the formation of the mixed layer helps maintain phytoplankton in the euphotic zone. Phytoplankton can then bloom because of high nutrient concentration and light availability. Then, phytoplankton becomes limited by nutrient availability and the bloom reduces. Coupling to the wave model consistently reduces stratification in April-May (Fig. 15) by increasing TKE, which leads to phytoplankton going through deeper cycles in the emergent mixed layer with longer time in deeper regions where light availability is lower. This delays phytoplankton bloom,
which happens 1 weeks to 2 weeks later with wave model coupling.

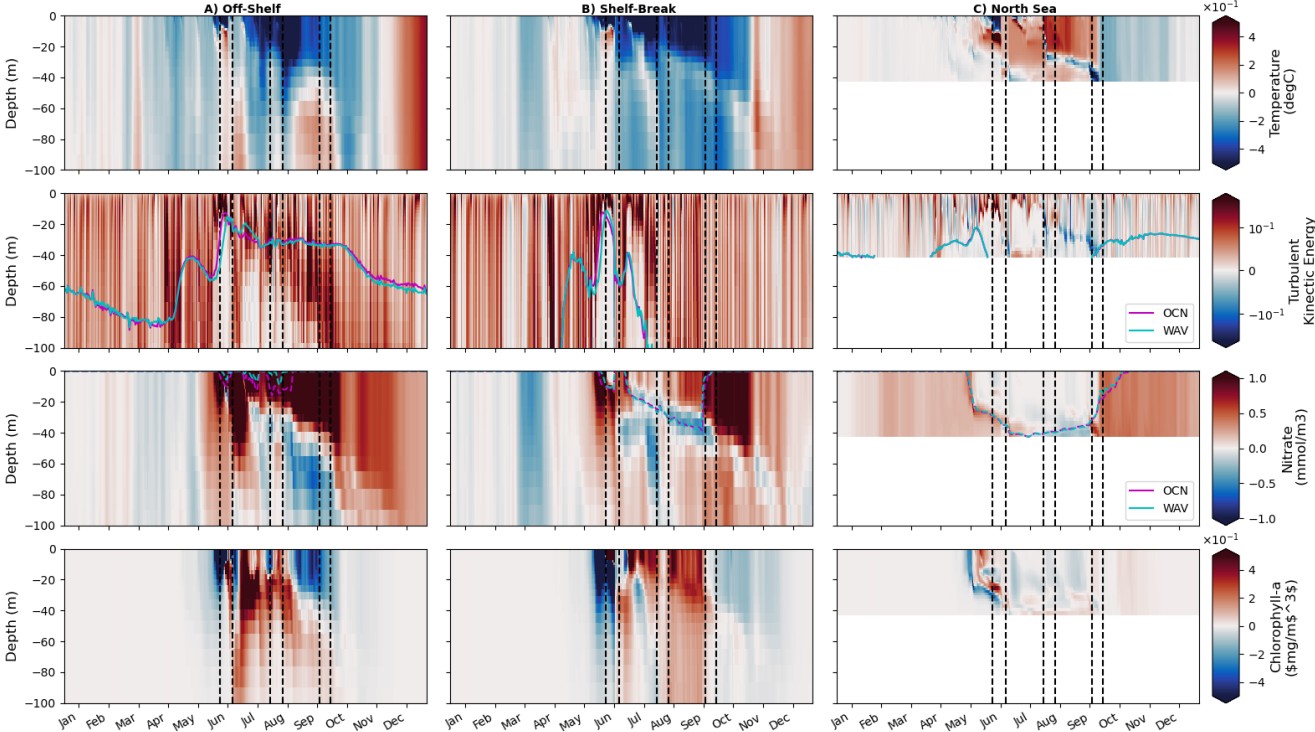

**Figure 14.** Hovmöller plots (depth/time) of WAV-OCN for potential temperature, turbulent kinetic energy, nitrate, chlorophyll and light availability for (left) off-shelf, (middle) shelf-break and (right) North Sea regions defined in Fig. 1. Magenta (OCN)/Cyan (WAV) time series show euphotic depth (1 % of surface light) in the second row and nutricline ($3\,\mu\mathrm{mol\,kg^{-1}}$ threshold) in the third.

In the North Sea, phytoplankton delay is still visible but the effect is less strong, as light is actually available throughout the column outside the phytoplankton bloom. Interestingly, each phytoplankton 'bloom' associated with the strong mixing events and termination of marine heatwaves is reduced by wave coupling, since the wave effect on these events is to reduce mixing by extracting atmospheric momentum by young waves. The apparent summer/autumn blooms after these events are not necessarily new growth, with increased surface chlorophyll the result of the mixing of the sub-surface chlorophyll maximum up from the nutricline to the surface, giving the appearance of a bloom.

## 6 Linking wave and phytoplankton activities

The relationship between wave activity and phytoplankton found in Fig 2 can be better understood from our analysis. Fig 5 showed that a mid-June storm tended to decrease phytoplankton activity off-shelf and increase it on the shelf break. The model was in agreement with this, but its response is stronger because this storm happens when chlorophyll levels are elevated in



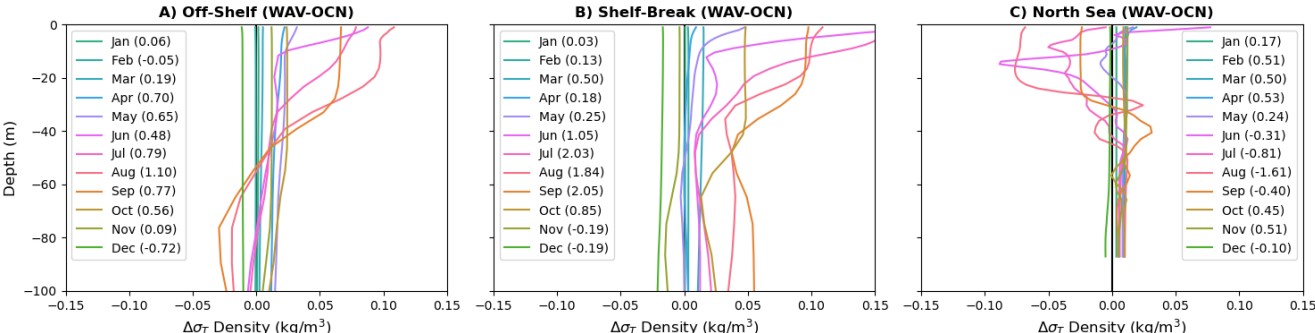

**Figure 15.** Difference in density ($kg/m3$) stratification profiles in the upper $150m$ throughout the year for the three regions. Density given in $\sigma_T (= \sigma - 1000)$ coordinates, with the total difference throughout the column for each month shown in parenthesis.

northern off-shelf regions in the model. Fig 13 shows the storm effect in the model is to mix the chlorophyll from $0\,\mathrm{m}$ to $20\,\mathrm{m}$ down to about $100\,\mathrm{m}$, hence the strong decrease in phytoplankton near the surface.


However, southwest of Ireland, both the observation and model have little activity before the storm, and increased activity after the storm. In this region, phytoplankton is limited by nutrients, which are brought nearer the surface by storm mixing because injected TKE reaches nutrient-rich water below the nutricline, as shown in Fig. 13 (shelf-break region).

In the North Sea, the model shows a stronger chlorophyll levels than the observations. However, for the July storm, the model agrees with the observations and doesn't increase in concentration. In this region, model profiles of chlorophyll and nutrients (Fig. 13) indicate that the bloom starts near the surface and then moves deeper, where nutrients are still available. In the model, the storm happens when the chlorophyll maximum is still at depth, bringing it back to the surface. In July, however, nutrients are depleted throughout the column and there is very little chlorophyll in the column, so the storm has no impact.
Therefore, the difference in the North Sea between model and observations in the June storm (Hector) is most likely due to the bloom being too late in the model.

In July, the off-shelf response of the model to the storm is too strong and suggests that the nutricline may be too shallow off-shelf in the model.


Finally, in the model, the first autumn storm (Ali/Bronagh) sees a small increase in chlorophyll in the North sea by increasing nutrient availability near the surface, but TKE does not quite reach the whole depth, and the chlorophyll concentration is underestimated. There are several feedbacks between the wave and atmosphere that are not represented by the models that could affect this, such as the change in surface roughness due to young waves.






## 7 Conclusion

This paper is the first usage of a km scale coupled wave-hydrodynamic-ecosystem model. We demonstrate the mechanistic impact of waves on shelf-seas ecosystems, and investigate how seasonal productivity can be altered by the sea state.

Our study identified regional and seasonal variations in the response of phytoplankton to wave activity: during the bloom (in March-April in the observations, April-June in the model), enhanced wave activity off shelf or near the shelf break induces a mix down of the surface bloom into deeper waters, and the chlorophyll concentration near the surface is reduced.

On-shelf, enhanced wave activity tends to generally increase chlorophyll activity near the surface from July to September, 450     likely because light is available throughout the water column, and wave mixing helps mix nutrients from rich to poor regions (either through advection of river plumes or vertically if the injection of TKE is deep enough and nutrients are still available near the bottom of the sea, which is possibly not the case in May-June (Fig 2).

Finally, phytoplankton activity also increases off shelf in the southwest part of the domain in June-September. The model's 455     response to storms in July and September suggests this is through bringing nutrients up from below the nutricline, when phytoplankton activity is very low.

This paper also investigated the effects of explicitly treating wave input to the ocean as a separate model coupled to the ocean with a 1.5km grid spacing. The main findings depend on the region of interest:

– off-shelf on the Atlantic side of the domain, explicit coupling with waves increases the injection of turbulent kinetic energy into the ocean across depth. This reduces stratification at the start of the growing season, meaning that phytoplankton is not trapped as close to the surface by an emergent stratification, and gets less light through deeper mixing cycles induced by stronger TKE. The bloom is therefore triggered later with the wave model. Total chlorophyll concentration and NPP are increased in summer, where stronger TKE injection leads to more intense phytoplankton secondary 465     blooms

– on the shelf-break, the conclusions are similar, but the increase in summer production is more pronounced than off-shelf

– on shelf, the changes are smaller for various reasons: outside bloom season, the euphotic depth is close to the actual shelf depth in the North Sea, so additional mixing does not affect the phytoplankton bloom as much. Second, the North Sea is sheltered by the British Isles and has younger waves than the Atlantic side. Young waves extract momentum 470     from the atmosphere before it is passed to the ocean, and reduce the amount of TKE in the ocean, this is clearly visible during summer storms, when marine heatwave stratification breaks: the mixing is less strong with the wave model and the various events of elevated chlorophyll concentration following these mixing events are less intense.



In line with previous findings, this study shows that coupling with a wave model improves the quality of the hydrodynamical model (Berthou et al., 2025b). However, it deteriorates phytoplankton activity: ERSEM has always been developed in a standalone ocean framework and may require further developments when used in a coupled system.

Future work will include comparing these simulations with a fully coupled simulation which includes a regional configuration of the Met Office atmosphere and land system: the physics only configuration is documented in a companion paper (Berthou et al., 2025a). In the present work, the ecosystem model does not feedback on the physical system: future work will include chlorophyll feedback on light penetration in the physical ocean building on Skákala et al. (2020), and sending surface chlorophyll back to the atmosphere for the calculation of surface albedo. Rivers will also get coupled to the ocean, and variable nutrient concentration eventually added too.

A fully coupled system will enable the study of compound events across land and marine environments in a changing climate. Representing the ocean colour feedback on the ocean and atmosphere was indeed recently shown to be important for climate projections in the Mediterranean, with resulting SST changes up to $1^{o}$C (Zhang et al., 2025).

**Code and Data Availability**

All scripts used to generate the biogeochemical input data are provided here: https://github.com/dalepartridge/AMM15_BGC_setup. The satellite data is available through OC-CCI (Sathyendranath et al., 2019)(Sathyendranath et al., 2023). The Met Office wave regional reanalysis is available from Copernicus: https://doi.org/10.48670/moi-00060. Given the size of 1-year of model output of this model, the data is stored locally and is available upon request by the authors.

**Author Contribution**

RM conducted analysis of observations vs wave data. DP set up and ran the model simulations with support from JMC and SB. Model analysis was conducted by DP and SB. Comparison between the model and EN4 profiles was done by JR. JC performed a literature review and biogeochemistry expertise. LB provided significant insight and expertise on wave processes. HL proposed and initiated the project. DP led the writing of the manuscript, with contributions to the writing and editing from all authors.

**Competing Interests**

The authors declare that they have no conflict of interest.



## 500 Acknowledgments

Thanks to ICDC, CEN, University of Hamburg for data support with NSBC validation data. DP, RM and JC acknowledge funding from UK Natural Environment Research Council's National Capability Long-term Single Centre Science Programme, Climate Linked Atlantic Sector Science (Grant Number: NE/R015953/1) and UK National Capability project FOCUS NE/X006271/1 (R.M.). DP, RM and JC also acknowledge funding from the NECCTON project, which received funding from Horizon Europe RIA under Grant Number 101081273. PML participants in NECCTON were supported by UKRI grant number 10042851. JR acknowledges support from the UK Marine and Climate Advisory Service (MCAS), part of the UK government's Earth Observation investment https://www.gov.uk/government/publications/earth-observation-investment/projects-in-receipt-of-funding.

This study has been conducted using E.U. Copernicus Marine Service Information; https://doi.org/10.48670/moi-00015



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
