# Peer review of "Impact of waves on phytoplankton activity on the Northwest European Shelf: insights from observations and km-scale coupled models"

_EGUsphere, 2025_

## Author Response (AR1)

**Response to Reviewer #1**

Thank you for your comments on the manuscript. They have helped us to revise the manuscript to become much clearer than the original submission. In particular we have created clearer distinctions between what is included in the two wave models and restructured the results sections to give more clarity and detailed discussion. In response to the specific comments:

In the introduction section, it is unclear from the literature review here what specifically did the authors mean by wave-induced mixing. Breaking surface wave (as only briefly mentioned by the authors) is certainly one important aspect. But there is literature on other non-breaking wave-induced processes that may enhance turbulent mixing, such as Langmuir turbulence (e.g., Belcher et al., 2012). It would be helpful to discuss more on what wave-related processes are relevant in this problem (processes that could potentially affect phytoplankton activities) and what processes were considered here in this study and why.

We agree that other wave related processes could be important, this part of the introduction has been entirely re-written to acknowledge the much more complex processes by which waves influence ocean mixing. We also made the description of what is included in the wave-ocean coupled model (S3.2) clearer with more details.

The introduction of the ocean-only simulation and ocean-wave coupled simulation is confusing. It seems that the enhanced surface mixing by surface wave-breaking and wave-modified sea surface roughness are included in both? Or did I misunderstand it? It would be helpful to elaborate more on how the effects of waves are included in the ocean-wave coupled simulations and to clarify what are the essential differences between the two simulations.

Yes, wave effects are taken into account in the ocean model, but calculated assuming wind/waves are in equilibrium in the ocean-only model. We have extended the description of the wave-ocean coupling to explain better how the terms are changed when including coupling from the wave model.

The authors attributed the reduced TKE during periods of the mixing events in the on-shelf region to the presence of young waves, whose growth reduces the transfer of momentum from the atmosphere to the ocean (if I understand it correctly). While the growth of waves certainly changes the air-sea momentum flux, I'm not sure this sufficiently explains the reduced TKE seen in Fig 14. It would be helpful to elaborate more on how this reduced TKE happened and if there are alternative explanations.

Thank you for highlighting the lack of demonstration of this point. We added a figure showing TKE averaged over the first 20m of the water column during the three summer storms and the corresponding Charnock parameter, which is a good indicator of wave

growth. The new figure shows well that sheltered seas receive less momentum during storms than the open ocean. In the regions of high Charnock parameter, energy is being used to grow waves. While in areas where the waves are older, and the seas are mode well developed, the energy from the atmosphere is passed directly to the ocean.

The organization of the results in Sections 4-6 may benefit from a careful reorganization and revision. Wave effects were discussed in all three sections, but the emphases were not clearly differentiated, which may cause confusion. Some discussions seem redundant while others seem insufficient. For example, comparison with observations showed delayed bloom in both simulations. This model bias could lead to different responses of phytoplankton activity to storms in models versus observations. How such model bias could potentially affect the impact of waves before and after the passage of the three storms is a point that probably deserves more discussion. And given the sensitivity of the response of phytoplankton activity to storms to the bloom onset time, some more discussion on the robustness of the results shown in the study may also be helpful.

Thank you for your comments on the clarity of the results. We have reorganised the sections to create more distinction and added/removed details to make everything clearer. We have removed some of the observation comparison to keep the focus on the impact of the additional model mechanisms, and the storms are now discussed in their own section to expand the discussion.

Specific comments

L69: How is this done implicitly? Changed the word implicit to parameterised along with a more detailed explanation in section 3.2

L70: There seems to be a gap in the literature review linking a better representation of Stokes drift to enhanced mixing. It is unlikely that the Stokes drift enhances the ocean mixing, we have made this clearer in the methods section: it is more likely the fact that the surface stress is now calculated by the wave model, rather than the ocean model. We have also included a new figure to address 3., and which shows that mixing is more dependent on wave growth/decay in the coupled system.

L77: It would be helpful to elaborate more on what kind of wave coupling is considered in this study. Rephrased, along with an expanded explanation of the wave model in section 3.2

L82-83: It would be helpful to be specific on what are the "implicit and explicit representation of wave mixing effects". Implicit waves now called parameterised waves to better reflect what is happening. Definitions have been improved and stated in section 3.2 to make it clearer

L96: first "section 5" -> "section 4"? Fixed

L131-134: Such correlation could also be a result of the influence of wind, available sunlight etc that correlate with wave energy? We agree with the reviewer, we added "Wave energy is likely not the sole causal factor in this correlation: increased storminess will also decrease available sunlight."

Section 2.3: Does the conclusion generalize to other years? We added "Indeed, some years have uninterrupted summer stratification (e.g. 2017, used in \cite{lewis2019}): we choose a year in which wave activity was enhanced in spring and had intense summer wave activity events able to disrupt the ocean stratification."

L147: Is there a starting point of an exponential growth? Its unclear what the reviewer means by this. The mathematical definition we are using is given in the following sentence.

L152: But how to deal with the different behaviors in the model and satellite data? Is there a way to identify occasions when the algorithm fails? Algorithm is not suited to use with satellite data as it works best with a full time series. We decided to removed the image of the bloom onset for satellite as it detracts from the explanations of the mechanistic processes we are trying to demonstrate

L166-167: How to attribute the changes in chlorophyll to wave activities? It is unclear from Fig. 5 whether wave activities have an impact on the changes of concentration of chlorophyll. This figure mainly shows that wave activity happened in spring and summer of 2018, motivating the use of this particular year to study the relationship, which will be done in the rest of the article. We removed the suggestion that the figure was demonstrating a relationship, which was wrong, and re-wrote the paragraph as such:

"The observations confirm 2018 to be an interesting year to study the relationship between wave activity and the concentration of chlorophyll. Indeed, some years have uninterrupted summer stratification (e.g. 2017, used in Lewis et al. (2019)): we choose a year in which wave activity was enhanced in spring and had intense summer wave activity events able to disrupt the ocean stratification. In the remainder of the article, we focus on how intense periods of wave activity affected the concentration of chlorophyll, and how explicitly coupling an ocean model to a wave model impacts model results."

L182: How is a high resolution helpful to the processes discussed in this study? Clarified and citations added

L190: "absence" -> "presence"? — the introduction of wave effects in the model is confusing. Rewritten this part to make clearer

L192: Langmuir turbulence is known to enhance vertical mixing significantly more than wave-breaking (e.g., Belcher et al., 2012). Why not try something like the model by Harcourt (2013, 2015)? Langmuir turbulence is not currently configured as a coupling

between NEMO and WAVEWATCH III due to limitations of GLS mixing scheme used in the operational configuration .

L190-194: It would be helpful to discuss a bit on why these wave effects were considered and why these parameterizations were chosen. There are other parameterizations for both wave-induced mixing and wave-modulated momentum flux.

This study was constrained by the existing operational configurations, both for the parameterised waves in the NEMO-only run and for the coupling between NEMO and WAVEWATCH III. Making changes to the codebase was outside the scope of this study.

L219-220: It is very confusing to have wave-breaking effects in the ocean only system.

Sections 3.1 and 3.2 have been rewritten to better explain the implementation of waves in both simulations

L211-220: This is very confusing given the introduction in L189-194. Not sure if I understand it correctly, but it seems that some of the wave effects are included in the ocean-only system? It would be helpful to reorganize the description of the ocean-only model and wave-ocean coupled model to clarify on what specifically do the authors mean by the wave effects in this study. As above

L230: Missing section number? Removed the bold text as it was unnecessary

L232: Please define "AMM7" Defined at the start of the section

L233: Is there a better option than extrapolation from nearest neighbor? Why there was a mismatch between the domain of these two configurations? How would such extrapolation in the initial condition affect the results? Nearest neighbour was used as the spin-up negates any impact in the runs. This has been made clearer in the text.

The AMM15 domain was defined by a larger consortium to match up with an atmospheric model domain, as such it is outside our control to define the extents

L238: "constant value" of what? They are set to a near-zero value typical for the variable in winter. Text has been amended to clarify this

L263: "stratifiies" -> "stratifies" Fixed

L266-269: How would this warm bias affect the results reported here? The warm bias would definitely have an impact on the phytoplankton bloom timings and strength, making it difficult to compare observations and model runs. The warm bias was consistent between the two model runs which still allows us to examine processes and drivers when explicit wave coupling is included. Added a comment in the conclusions to reflect this.

L274-276: These statements seem to conflict with numbers in Table 1. For example, spatial correlation for chlorophyll is low in Table 1. We agree that the numbers highlight

differences between the model and observations, the initial reasoning behind including them was to demonstrate that they largely followed the same patterns even if there were difference. From the reviewer comments it seems this was confusing and so we have removed some of the comparison between model and observations to keep the focus on the mechanistic drivers from the twin experiment.

L285: Consistent only on the seasonal time scale (which should be expected)? But there seems to be large differences in the variation of temperature between models and observations? As above

L287: Maybe labeling the marine heatwave events in the figure will be helpful. Thank you for the suggestion, these have been added along with the storm timings.

L290-291: I don't understand this sentence. This sentence related to comparing the bloom onset between the observations and the models. We have removed this to avoid confusion as described above.

L297: There seems to be large variability of both simulated Chl-a and NPP when they started to increase in Feb-Mar. The difference between OCN and WAV in Fig. 8 does not seem to be significant to me. It might be helpful to label the bloom onset time in Fig. 8 to give readers a reference on what does it really mean by the onset delay shown in Fig 9. It also seems to me that this onset delay may strongly depend on the definition of the bloom onset. So it would be helpful to discuss how robust are the results in Fig. 9. The location of the L4 station, where these observations are taken is not an area where waves will have a significant impact on the results. This has been made clearer in the text.

Fig. 8c: Missing Chl-a OBS data after April? Fixed

L325-326: Is there a way to distinguish the effect of a later bloom onset (WAV vs. OCN) and the effect of waves during a storm in these results? We have added a more detailed discussion on the impact of storms, although it would be difficult to isolate the effects

L331: What do the authors mean by "while accounting for biases in the models relative to observations"? Changed to 'independent of the biases relative to the observations'

L357: It might be helpful to first introduce these two figures before discussing the results shown in these figures. Added text to introduce the figures

L373-377: It would be helpful if the authors could elaborate more on why TKE is reduced when wave effects are included in this case. What are the key differences between locally generated wind waves and remotely generated swells in affecting the TKE in this region? We added in figure 15 and some discussion in an expanded section covering the storm impact

Fig 13 caption: "North Sea" -> "on-shelf" to be consistent with phrases used in other places? Changed

L409-410: Do the authors mean that during these periods atmospheric momentum is used to grow waves rather than generating mixing in the ocean? Do those waves result in stronger mixing later?

Answered below under L410-412

L410-412: Please rephrase this sentence. Rephrased

Section 6: I don't see why it is necessary to separate the discussion in this section from the discussion of Figs 13-14 in the previous section. This discussion has been added to the previous section

L469-470: I don't understand this statement. Waves cannot keep growing indefinitely. At some point they will deposit momentum to the ocean (perhaps at a different location)?

We have added the following passages into the manuscript in the relevant places:

"Interestingly, each phytoplankton 'bloom' associated with the strong mixing events and termination of marine heatwaves is reduced by wave coupling. In the coupled case, momentum from the atmosphere is used to grow young waves. This delays the transfer of energy from atmosphere to ocean. In the other case, when waves are not explicitly included, the energy can abruptly mix the water column without this intermediate step."

"Young waves extract momentum from the atmosphere before it is passed to the ocean, and delay the transfer of TKE in the ocean, this is  visible during summer storms, when marine heatwave stratification breaks. In the WAV case, the waves delay transfer of momentum and spread the event over a longer time acting to 'flatten the curve'. Thus the mixing event is less sharp and intense, and the elevated chlorophyll concentrations less peaked."

L473-474: I don't think sufficient evidence is provided to support these conclusions — the differences between WAV and OCN are much smaller than the difference between models and satellite observations? We have clarified the conclusions in the context of the bias in the system

**Response to Reviewer #2**

We thank the reviewer for their comments. As part of the revisions we have added further discussion to the limitations of the model and highlighted both the issues that prohibit an in depth comparison with observations and the reason why the twin experiment still provides useful mechanistic understanding. The results section has been restructured to help make the explanations more cohesive and to give further discussion to the role of storms.

Specific comments:

Line 45: Can the spring–neap tidal signal be identified in the results? We did not look into the tidal signal as it was out of scope of the study.

Line 75: Is this study consistent with previous findings? This point should be addressed in the Discussion section. Added a line in the conclusions discussing this.

Line 130: The statement "the shelf break … is generally negative, except in August–September" is not entirely accurate, as the values are not statistically significant from May to July. Please consider revising for clarity and precision. Revised the text to say 'where significant'

Line 130: Could the authors provide a dynamical explanation for the contrasting positive and negative correlations observed north and south of the off-shelf region? This would help clarify the underlying physical processes.

In red areas from May-Aug we see more wave energy -> more chlorophyl (because the waves are mixing-up nutrients in the summer months, when the plankton are nutrient limited. Looking at fig 13a, we see the monthly stratification split into these 2 clusters. And looking at fig 11a (off-shelf), the strong nutricline is present in these months. In the blue areas, in the autumn+winter Sep-Mar, do we see the waves causing deeper mixing, so they switch to being light limited?

Figure 2: Using $p < 0.1$ as a significance threshold is more permissive than the conventional $p < 0.05$ standard. The authors may consider justifying this choice.

We relaxed the threshold as the data is sparse. We were able to draw the same conclusions when using $p<0.05$, but visually $p<0.1$ provide a clearer picture.

Figure 4: The blue color appears saturated in this figure, indicating that the bloom onset occurred earlier than April 11. I suggest adjusting the colorbar range accordingly. This image has been removed from the manuscript, as we felt it detracted from the message we want to get across.

Figure 5: Consider adding the storm track to the map to enhance visualization. The same suggestion applies to Figure 10. Where available, storm tracks have been added

Line 160: This paragraph attributes the differences before and after the storm to its impact; however, multiple processes could contribute to these changes and should be acknowledged in the text. We agree that multiple processes can be at play but have added a further plot and discussion to support the claim that it's the dominant process at the time.

Line 260: It is unclear whether the observations were taken during the same time period as the simulation. This has been made clearer

Line 275: The reported spatial correlation of –0.14% appears unreasonable for validation. Please verify this value and clarify how it was calculated. We have clarified in the text the bias issues and removed some of the observation comparison to focus on the mechanisms at play.

Line 290: I am concerned about the delayed bloom in the simulations compared to the observations. The delay attributed to wave effects appears relatively small compared to the overall bias with observations. What could be the possible reasons for the delayed bloom in the model? Has any tuning been attempted to reduce the biases in temperature, chlorophyll concentration, or bloom timing? As above

Figure 8: For chlorophyll, it is recommended to include the satellite-derived values in panels (c) for comparison. Added in to the plot

Line 310: The spatial patterns of bloom timing differ notably between the satellite data (Fig. 4) and the model results (Fig. 9). For instance, the observed later bloom onset south of England is not captured by the model. Additionally, the model shows an expansion of late bloom in the off-shelf region between 55–60°N, which is not evident in the satellite observations for the same area. As before, the focus is on the mechanisms between the twin runs. The satellite bloom onset plot has been removed to help keep the message clear

Line 325: A significant discrepancy in chlorophyll is evident between the satellite and model results on 2018-06-06. Moreover, the storm response differs between the two: in the simulations, chlorophyll decreases in regions with high concentrations and increases where concentrations are low. This inconsistency makes it challenging to identify a unified mechanism through which storms influence chlorophyll. The causes of these differing responses and their underlying mechanisms warrant further discussion. We have added a section explicitly focusing on the storm response in the model.

Figures 8, 11, and 12: Consider adding dashed lines to indicate the timing of storm events. Added to the plots

Line 390: The phrase "Chlorophyll is only increased" should be revised to "Chlorophyll mainly increases"). Consider revising the sentence for clarity: "Off-shelf, increased nutrient concentrations near the surface may be offset by reduced light availability due to a deeper mixed layer and enhanced vertical mixing that transports chlorophyll out of the euphotic zone." Also, please ensure consistent use of "off shelf" vs. "off-shelf" throughout the manuscript. Thank you for the suggestion, we have revised the sentence accordingly

Line 400: If reduced light availability were the main factor delaying the bloom, one might expect elevated chlorophyll concentrations either near the surface or below the euphotic layer due to enhanced vertical mixing. However, the results show a consistent decrease in chlorophyll throughout the water column before June. Could this indicate that temperature plays a more dominant role in controlling the timing of the bloom? The reviewer is correct to point out that reduced temperature will also be a factor in controlling the bloom. We have amended the text to reflect this.

---

## Author Response (AR2)

**Reviewer 1**

The authors have addressed my previous comments properly and the clarity of this manuscript is greatly improved. I only have some additional minor comments as in the following, mostly suggestions for clarification in the introduction.

Thank you again for your input, it has been really useful.

Specific comments:

L3: "waves" -> "surface waves"? Fixed

L9: "two-way coupled model" -> "two-way coupled ocean-wave model"? Fixed

L10, and following lines in the abstract: "wave mixing" -> "wave-driven mixing"? Changed

L69-70: Its effects, in particular in driving Langmuir turbulence and thus enhancing vertical mixing, is parameterized in ocean models. See, e.g., Li et al., 2019. Added a line to this affect

L79-80: I think the terminology is confusing here. Even with explicitly coupled wave model, the effect of surface waves on the mixing is still parameterized in an ocean model — wave phases are not resolved in ocean general circulation models. I think what the authors meant here is parameterized wave variables, not wave-induced mixing (which is always parameterized in the ocean model regardless of where the wave information comes from). Amended

L82: Two-way coupling itself does not induce enhanced vertical mixing — there has to be a vertical mixing parameterization that depends on wave variables provided by the two-way coupling. Changed 'induce enhanced' to 'enhance' to be more reflective

L85-86: As alluded here, there are two components of incorporating the effects of wave-driven mixing in an ocean model. (1) a turbulence parameterization scheme that depends on wave variables and (2) a source of wave information — from a wave model in the two-way coupled wave-ocean setup and some empirical relations in the ocean-only setup. So, the wave-driven mixing parameterization is as important as wave-ocean coupling, which should probably be discussed a bit more in this introduction. At least, it should be clarified what specific mechanism of wave-driven mixing is considered here. Is it wave breaking, Langmuir turbulence, or something else? The additional processes considered are detailed in section 3.2, we have added them into the introduction at line 102

L97: See comments above. I think the authors meant explicit representation of wave statistics here, not wave-driven mixing effects. Fixed

L102: It might be useful to make it clear in the introduction what specifically are these wave-induced processes? See above

L220-222: I think the depth integrated Stokes drift is also needed in addition to the surface Stokes drift in order to use the method in Breivik et al., (2016) to reconstruct the full Stokes drift profile? True, this paragraph focuses on what is added by the wave model that makes it different from the ocean only case rather than a full discussion of the schemes.

L239-241: Also Langmuir turbulence enhanced vertical mixing, which is more relevant to the GLS turbulence scheme here. See, e.g., Li et al., 2019. Noted and reference added

**Reviewer 2**

Major comments:
The revised manuscript addresses the previous comments and demonstrates notable improvements in the interpretation and discussion of the results. However, the current version suggests that the model may overestimate temperature and chlorophyll concentrations, potentially leading to an exaggerated assessment of wave and storm impacts on net primary production (e.g., comparison of Figure 4 and Figure 14). Clarifying this issue would strengthen the credibility of the study's conclusions regarding the biogeochemical effects of wave processes.

Thanks for the comment. In the text we acknowledge that the impact is larger in the model due to the overestimation, but there is enough to suggest that the patterns of the response are similar. The use of a twin experiment in this case allow us to draw conclusions about processes and effects if not the scale. We have added text into the conclusions to clarify this.

Detailed comments:
Figure 9: To better account for regional differences in mean concentrations, it would be helpful to plot the time series of the percentage change relative to the OCN case. This would also help to more clearly highlight the significance of wave effects. Due to low concentrations over the winter period the percentage change amplifies values during that time and detracts from the differences in the relevant months

Line 350: I would expect the warm bias in the model to lead to an earlier bloom onset. The bloom onset is dictated by many factors and BGC models often struggle to predict the timing in free running simulations. Previous studies that did not have a warm bias have also seen an onset later than observations suggest.

Line 355: This paragraph should be revised to after the removal of the observed bloom onset figure. Consider either including the figure in the supplementary materials or explicitly noting that the figure is not shown. Removed the reference to the removed figure

Figure 12: Although it is not the primary focus of this study, what might explain the higher winter temperatures with enhanced mixing shown in the WAV case? In the autumn and winter the surface of the ocean cools down. Enhanced mixing brings water that is now warmer than the upper layers towards the surface, effectively dampening the cooling of the top 100m of the ocean. It is the opposite of what happens in the spring when the surface is warming. The whole annual cycle of warming and cooling is dampened by enhanced mixing.

Figure 13: Should be "in the upper 100m" and this figure is not necessary in the main text of the manuscript. Fixed the caption. Whilst this figure could be removed we feel

demonstrating the monthly variation to the stratification difference between the two model runs highlights the impact.

Line 488: The increase in summer production is less pronounced than off-shelf. Fixed